

# On duality between Cosserat elasticity and fractons

**Andrey Gromov[1*] and Piotr Surówka[2†]**

**1** Brown Center for Theoretical Physics & Department of Physics,
Brown University, Providence, Rhode Island 02906
**2** Max-Planck-Institut für Physik komplexer Systeme,
Nöthnitzer Str. 38, 01187 Dresden, Germany

* andrey_gromov@brown.edu, † surowka@pks.mpg.de,

## Abstract

We present a dual formulation of the Cosserat theory of elasticity. In this theory a local element of an elastic body is described in terms of local displacement and local orientation. Upon the duality transformation these degrees of freedom map onto a coupled theory of a $U(1)$ vector-valued one-form gauge field and an ordinary $U(1)$ gauge field. We discuss the degrees of freedom in the corresponding gauge theories, relation to symmetric tensor gauge theories, the defect matter and coupling to the curved space.



## 1  Introduction

Duality is a powerful tool that allows one to access non-perturbative physics of interacting systems. Recently there was a resurgence of interest in various dualities in a quantum field theory [1–3]. The most well-known example of duality in quantum field theory is the boson-vortex duality [4–7], which maps a superfluid in 2+1 dimensions to an Abelian Maxwell theory. Upon this transformation the superfluid vortices are mapped to matter, charged under the dual gauge field.

In a parallel development a duality transformation for a quantum theory of elasticity was constructed [8–11]. It turns out that the dual theory is an Abelian gauge theory of (symmetric) tensor gauge field. Such theories have recently emerged in condensed matter physics in the study of algebraic spin liquids [12–14], and, later, of gapless fracton phases [15–17]. Symmetric tensor gauge theories couple to an unusual type of matter. The dynamics of such matter is restricted by a set of global conservation laws, that preserve not only the total charge of the system, but also various multipole moments of the charge density. These conservation laws lead to partial, or complete, immobility of charged quasiparticles [16, 18, 19]. The phenomenon of restricted mobility of excitations has recently attracted attention in the study of topological phases of matter, spin liquids and self-correcting quantum memory. Namely, a new type of topological order, *fracton* order was found and established [20–35]. Fracton systems exhibit certain similarities to traditional topologically ordered phases, namely topological[1] groundstate degeneracy on a torus and robustness to local perturbations. At the same time, fracton phases are quite unusual in that the topological degeneracy depends exponentially on the system size and on the presence of topologically non-trivial lattice defects, such as disclinations.

The relationship between fracton phases of matter and elasticity has been noticed by several authors [36–42]. While formal details are different among these works, the essential observation is quite simple. Crystalline defects, such as dislocations and disclinations exhibit the phenomenon of restricted mobility. In particular, dislocations have to satisfy the so-called glide constraint, which forces them to move along their Burgers vector, provided that total number of lattice sites is conserved, while disclinations cannot move without creating dislocations. This parallels the phenomena in the physics of type-I gapless fracton phases where certain fractons can only move via creating other fractons.

One way to access the physics of crystalline defects is to utilize the duality transformation, which maps a theory of elasticity onto a tensor gauge theory coupled to crystalline defect matter. In this work, we consider a generalized theory of elasticity, known as micropolar or Cosserat elasticity [43–45]. In Cosserat elasticity, the elastic medium is equipped with a microstructure, which has a microscopic origin. Accordingly, a local volume element is described (in $2+1$ dimensions) by a displacement vector $u^i$ and a local orientation $\theta$. In thermal equilibrium (or in the ground state, at $T = 0$) both of these fields constant in space. The theory is assumed to have a global translational and rotational symmetry (although the latter requirement can be relaxed). The displacement field is generally gapless, while the orientation field is

---

[1]That is, present in the absence of any symmetry.

generally gapped. We construct a dual theory via solving the constraints imposed by the conservation of momentum and (non-)conservation of the angular momentum. The dual theory contains a general tensor gauge field (more precisely, a vector-valued one-form gauge field) and a real $U(1)$ one-form gauge field. This gauge theory contains two gapless and one gapped degree of freedom. The crystalline defects, which are singularities in the displacement and rotation fields, naturally couple to these gauge fields.

The manuscript is organized as follows. In Section 2 we provide an introduction to the dual formulation of the theory of elasticity. This section contains no new results, but is used to fix the notations and to make the manuscript self-contained. In Section 3 we describe the duality transformation for Cosserat elasticity. In Section 4 we present our conclusions. Appendix A is devoted to the inversion of elasticity coefficients and Appendix B to the Stückelberg mechanism of the massive mode in the gauge theory dual to Cosserat elasticity.

## 2 Symmetric elasticity

This section serves as an introduction to the elastic dualities. The results contained here can be found in Refs. [36–38, 40, 41].

### 2.1 Symmetric elasticity

We start with an introduction to the ordinary theory of elasticity. The fundamental assumption in the traditional, "symmetric" elasticity is the lack of local structure of the elastic medium. All deformations, both elastic (phonons) and plastic (dislocation and disclination defects) are described using the displacement field $u^i$ [46]. Smooth variations of $u^i$ correspond to the smooth distortions of the lattice, whereas singular configurations of $u^i$ correspond to the lattice defects. We will assume complete rotational invariance.

In symmetric elasticity we introduce the symmetric strain tensor $u_{ij} = \partial_i u_j + \partial_j u_i$. The action (or free energy at finite temperature) is assumed to depend *only* on $u_{ij}$.

$$S[u^i] = \int dt \, d^2x \left[ \dot{u}_i \dot{u}_i - C^{ijkl} u_{ij} u_{kl} \right], \tag{1}$$

where $C^{ijkl}$ is a tensor of elastic moduli. In two spatial dimensions it has two independent components and encodes shear and bulk elastic moduli. The summation over repeated indices is assumed. Note that, in the definition of the elastic coefficients, we do not follow the standard convention for symmetric elasticity [46]. Our choice will facilitate the comparison of the results of this section to Cosserat elasticity. The equation of motion takes the form of a conservation law for momentum density, $P_i$. We introduce the momentum density as $T^{i0} = P^i$ and $T^{ij}$ is the stress tensor. Then the conservation of momentum takes the form

$$\dot{P}^i + \partial_j T^{ij} = 0 \qquad \Longleftrightarrow \qquad \partial_\mu T^{i\mu} = 0. \tag{2}$$

The stress tensor is given by

$$T^{ij} = C^{ijkl} u_{kl}. \tag{3}$$

The partition function for elasticity reads

$$Z = \int Du^i e^{iS[u^i]}. \tag{4}$$

Next we reformulate the partition function in terms of the dual variables by essentially performing a Legendre transformation. Using the Hubbard-Stratonovich trick the action is

brought to the following form

$$S[P^i, T^{ij}, u^i] = \int dt d^2x \left[ P_i P^i + C^{-1}_{ijkl} T^{ij} T^{kl} + u_i (\partial_\mu T^{i\mu}) \right].$$ (5)

We discuss the inversion of $C^{-1}_{ijkl}$ in Appendix A. The resulting partition function is given by

$$Z = \int DP^i DT^{ij} Du^i e^{iS[P^i, T^{ij}, u^i]} = \int DP^i DT^{ij} Du^i_{\text{sing}} e^{iS[P^i, T^{ij}, u^i_{\text{sing}}]} \delta\left(\partial_\mu T^{i\mu}\right),$$ (6)

where in the second line the integral over (the smooth part of) $u^i$ was taken. To resolve the $\delta$-function we introduce the dual variables.

## 2.2 Duality

We are going to resolve the constraint $\delta\left(\partial_\mu T^{i\mu}\right)$ by introducing a tensor gauge field

$$T^{i\mu} = \epsilon^{\mu\nu\rho} \partial_\nu A^i_\rho.$$ (7)

The "vector potential" in this case is a vector-valued one form $A^i = A^i_\mu dx^\mu$. A representation of the stress tensor in terms of a vector potential is not unique. The formulation contains the gauge redundancy of the stress tensor

$$\delta A^i_\mu = \partial_\mu \alpha^i.$$ (8)

In components we have

$$P^i = \epsilon^{kl} \partial_k A^i_l, \qquad T^{ij} = \epsilon^{jk}(-\partial_0 A^i_k + \partial_k \Phi^i).$$ (9)

We introduce a notation $\Phi^i = A^i_0$. It is convenient to define the generalized electric and magnetic fields

$$B^i = \epsilon^{kl} \partial_k A^i_l, \qquad E^i_j = \epsilon^i{}_k(-\partial_0 A^k_j + \partial_j \Phi^k).$$ (10)

Thus the momentum and stress tensor map to the vector magnetic field and tensor electric field

$$P^i = B^i, \qquad T^{ij} = \epsilon_i{}^k \epsilon_j{}^l E^{kl}.$$ (11)

It is important that the index $i$ is *spatial* in nature. This means that the components of the one-form $A^i_\mu$ are spatial vectors. Under (spatial) coordiante transformations $A^i_\mu$ transforms as $(1,1)$ tensor

$$\delta A^i_\mu = \xi^k \partial_k A^i_\mu + A^i_j \partial_\mu \xi^j - A^j_\mu \partial_j \xi^i.$$ (12)

The antisymmetric part of the stress tensor is given by antisymmetric part of the tensor electric field $E^i_j$

$$T_{\text{odd}} = \epsilon_i{}^j T^i_j = \epsilon_i{}^j E^i_j.$$ (13)

In symmetric elasticity the stress tensor is symmetric, consequently there is a further local constraint

$$\epsilon_i{}^j E^i_j = 0 \qquad \Longleftrightarrow \qquad E_{ij} = E_{ji}.$$ (14)

This constraint can be written in terms of a vector potential

$$\partial_0 (A_{ij} - A_{ji}) = \partial_j \Phi_i - \partial_i \Phi_j.$$ (15)

This is solved by a symmetric vector potential and curl-free $\Phi^i$. That is,

$$A_{ij} - A_{ji} = 0, \qquad \epsilon^{ij} \partial_i \Phi_j = 0.$$ (16)

The first constraint is not gauge invariant. Indeed, applying a gauge transformation to the first constraint and demanding the invariance we find a constraint on the gauge freedom

$$\partial_i \alpha_j - \partial_j \alpha_i = 2\epsilon^{ij}\partial_i \alpha_j = 0. \tag{17}$$

Thus, $\alpha_i$ is curl-free, which, in two dimensions, implies that $\alpha_i$ is a total gradient $\alpha_i = \partial_i \alpha$. The second constraint implies that $\Phi_i$ is also a total gradient $\Phi_i = \partial_i \Phi$. Thus, the symmetric tensor gauge field $A_{ij}$ and the scalar potential $\Phi$ transform as

$$\delta A_{ij} = \partial_i \partial_j \alpha, \qquad \delta\Phi = \dot{\alpha}. \tag{18}$$

The action (5) takes the form

$$S[B^i, E_{ij}] = \int dt\, d^2x \left[ \tilde{C}_{ijkl}^{-1} E^{ij} E^{kl} + B^i B_i + \rho^i \Phi_i + A_{ij} J^{ij} \right] \tag{19}$$

$$= \int dt\, d^2x \left[ \tilde{C}_{ijkl}^{-1} E^{ij} E^{kl} + B^i B_i + \rho \Phi + A_{ij} J^{ij} \right], \tag{20}$$

where $\tilde{C}_{ijkl}^{-1} = \epsilon^{ii'}\epsilon^{jj'}\epsilon^{kk'}\epsilon^{ll'} C_{i'j'k'l'}^{-1}$ and $C_{ijkl}^{-1}$ is the inverse tensor of elastic moduli (see Appendix A). In the first line we use the vector charge formulation and in the second line we use the scalar charge formulation. We have also introduced the sources $\rho$ and $J^{ij}$, which, as we will argue in the next section, are mapped to the crystalline defects. Finally, we observe that the canonical momentum conjugate to $A_{ij}$ is

$$\frac{\delta S}{\delta \dot{A}_{ij}} = 2\tilde{C}_{ijkl}^{-1} E^{kl} = \Pi_{ij}. \tag{21}$$

The Gauss law associated with the gauge symmetry (46) takes the form (which is particularly simple in terms of the canonical momentum)

$$2\partial_j (C_{ijkl}^{-1} E^{kl}) = \partial_j \Pi^{ij} = \rho_i, \tag{22}$$

where $\rho_i$ is the vortex density. This is the Gauss law for the "vector charge" theory. After imposing the symmetry of the stress tensor we find the reduced Gauss law

$$\partial_i \partial_j \Pi^{ij} = \rho, \tag{23}$$

which is the Gauss law for the "scalar charge" theory. Moreover, taking the divergence of (22) we find that the "vector charge" $\rho_i$ is the first moment of the scalar charge $\rho$

$$\partial^i \rho_i = \rho \qquad \Rightarrow \qquad \rho_i = \int x_i \rho. \tag{24}$$

The gauge symmetry then implies a continuity equation for the symmetric tensor current

$$\dot{\rho} + \partial_i \partial_j J^{ij} = 0. \tag{25}$$

## 2.3 Mapping to defects

The densities of crystalline defects are given by [8, 47–49]

$$\rho_{\text{vac}} = \partial_i u^i, \quad \rho_{\text{disl}}^i = \epsilon^{kl}\partial_k \partial_l u_{\text{sing}}^i, \quad \rho_{\text{disc}} = \epsilon^{ij}\partial_i \partial_j \varphi, \tag{26}$$

where $\rho_{\text{vac}}, \rho^i_{\text{disl}}, \rho_{\text{disc}}$ are the densities of vacancies, dislocations and disclinations correspondingly. We have introduced a shorthand for the curl of the displacement vector

$$\varphi = \frac{1}{2}\epsilon^i{}_j \partial_i u^j_{\text{sing}}. \tag{27}$$

Under a local rotation by an angle $\psi$, $u'^i = R^i{}_j(\psi)u^j$, the angle $\varphi$ is shifted as follows

$$\varphi' = \varphi + \psi. \tag{28}$$

In symmetric elasticity the existence of smooth, globally defined displacement field, $u^i$, is guaranteed by the integrability conditions [8]

$$\rho^i_{\text{disl}} = 0, \qquad \rho_{\text{disc}} = 0, \tag{29}$$

which are equivalent to the absence of dislocation and disclination defects.

We now show that upon the duality transformation the defect densities map onto the densities of dual charges. In particular, we need to show that the density of disclinations maps on the scalar charge density. To do this we decompose the displacement vector in (19) as $u^i = u^i_{\text{reg}} + u^i_{\text{sing}}$. Integrating over the regular part of $u^i$ leads to the conservation of the momentum constraint.

The singular part of $u^i$ then couples as follows

$$\delta S_{\text{vortex}} = \int dt d^2x \left[ u^i_{\text{sing}} \partial_\mu T^{i\mu} \right] = \int dt d^2x \left[ \rho^i \Phi_i + J^{ij} A_{ij} \right] = \int dt d^2x \left[ \rho \Phi + J^{ij} A_{ij} \right], \tag{30}$$

where $\rho^i = \epsilon^i{}_j \epsilon^{kl} \partial_k \partial_l u^j_{\text{sing}}$ and $J^{ij} = \epsilon^i{}_n \epsilon^{\mu\nu k} \partial_\mu \partial_\nu u^n_{\text{sing}}$. Thus we find that dual charge density maps onto rotated dislocation density

$$\rho^i \qquad \Longleftrightarrow \qquad \epsilon^i{}_j \rho^j_{\text{disl}}, \tag{31}$$

which implies that elementary vector charges $\rho^j$ are dislocations with the Burgers vector $b^i = \epsilon^i{}_j \rho^j$. The scalar charge density maps onto the density of disclinations by the virtue of the following elementary identities

$$\epsilon_{ij} \partial^i \rho^j_{\text{disl}} = \rho_{\text{disc}}, \qquad \partial_i \rho^i = \rho. \tag{32}$$

## 2.4 Glide constraint

The dynamics turns out to be further constrained if the number of point defects (vacancies and interstitials) is not allowed to fluctuate. Indeed, consider the following moment

$$Q_2 = \int x_i x^i \rho = \int E_i{}^i = \int \partial_i u^i = \int \rho_{\text{vac}}. \tag{33}$$

Thus, if the total number of vacancies, $\int \rho_{\text{vac}}$, remains constant in time, then so does $Q_2$. Conservation of $Q_2$, on the other hand, implies that the dipoles can only move perpendicular to their dipole moment. The motion along the dipole moment changes $Q_2$ because it requires adding an interstitial or a vacancy. This corresponds to a well known fact in the theory of elasticity: the dislocations can only move along their Burgers vector.

This completes our introduction to the duality transformation of 2D quantum elasticity.

## 3 Duality for Cosserat elasticity

### 3.1 Cosserat elasticity

Cosserat elasticity is a generalization of the symmetric elasticity, which considers an elastic medium with a microstructure. There are various mechanisms for such a structure to be generated, such as multiple atoms in the unit cell or higher gradient effects in the symmetric elasticity. The major assumption is that local elements of the medium can transfer both forces and *torques* to nearby elements of the medium. In the realm of classical elasticity effects associated to Cosserat elasticity were recently observed in the context of metamaterials [50, 51], where the lattice constant of the medium is mesoscopic and the building blocks are chiral, thus allowing the transfer of angular momentum through shear stresses.

To account for the microstructure, in addition to the displacement vector $u^i$, we introduce a local angle that describes a local orientation of the medium, $\theta$. Because of the local torques the stress tensor in the medium is no longer symmetric. Formally, we introduce a non-symmetric strain tensor and a rotation vector

$$\gamma_{ij} = \partial_i u_j - \epsilon_{ij}\theta, \qquad \tau_i = \partial_i\theta. \tag{34}$$

The anti-symmetric part of the strain tensor now takes form

$$\frac{1}{2}\epsilon^{ij}\gamma_{ij} = \frac{1}{2}\epsilon^{ij}\partial_i u_j - \theta = \varphi - \theta. \tag{35}$$

Upon choosing $\theta = 0$ and picking a preferred frame of reference, that decouples the antisymmetric part stemming from non-zero orbital angular momentum, the Cosserat theory reduces to the symmetric elasticity. $\varphi = \theta$ corresponds to a special limit of Cosserat elasticity, in which the local rotation or spin is completely frozen, known as the couple-stress theory [45, 52]. In general Cosserat theory, the field $\theta$ is completely independent of $u^i$.

The action is a functional of $\theta$ and $u^i$. It reads

$$S[u^i, \theta] = \int dt d^2x \left[ \dot\theta\dot\theta + \dot u^i\dot u^i - C^{ijkl}\gamma_{ij}\gamma_{kl} + \zeta\tau_i\tau^i \right], \tag{36}$$

where $C^{ijkl}$ and $\zeta$ correspond to elastic coefficients in the theory. The Hubbard-Stratonovich transformations bring the partition function to the form

$$Z = \int Du\, D\theta\, DP\, DT\, DL\, e^{iS[u,\theta,P,T,L]}, \tag{37}$$

where

$$S = \int dt d^2x \left[ P_i P^i + (L_0)^2 + \zeta^{-1}L_i L^i + C^{-1}_{ijkl}T^{ij}T^{kl} + u_i\left(\partial_\mu T^{i\mu}\right) + \theta\left(\partial_\mu L^\mu - \epsilon^{ij}T_{ij}\right) \right]. \tag{38}$$

Integrating out (the smooth part of) $\theta$ and $u_i$ leads to the following constraints

$$\partial_\mu T^{i\mu} = 0, \qquad \partial_\mu L^\mu - \epsilon^{ij}T_{ij} = 0. \tag{39}$$

These constraints correspond to conservation laws of momentum and angular momentum. It is important to note that in the Cosserat theory the stress tensor is not symmetric. We will resolve the constraints by introducing gauge fields.

## 3.2 Duality

The constraints (39) are dealt with in two steps. First, we introduce the tensor gauge field to resolve the first constraint as before

$$\delta A^i_\mu = \partial_\mu \alpha^i. \tag{40}$$

In components we have

$$P^i = \epsilon^{kl} \partial_k A^i_l, \qquad T^{ij} = \epsilon^{jk}(-\partial_0 A^i_k + \partial_k \Phi^i). \tag{41}$$

We follow the notation of the previous section introducing $\Phi^i = A^i_0$ and the generalized electric and magnetic fields

$$B^i = \epsilon^{kl} \partial_k A^i_l, \qquad E^i_j = \epsilon^i{}_k(-\partial_0 A^k_j + \partial_j \Phi^k). \tag{42}$$

Thus momentum and stress tensor map onto the vector magnetic field and tensor electric field

$$P^i = B^i, \qquad T_{ij} = \epsilon_i{}^k \epsilon_j{}^l E^{kl}. \tag{43}$$

In the Cosserat theory the stress tensor is *not symmetric*. Therefore we cannot implement the same reduction of asymmetric components. Consequently, the Cosserat theory is *not* dual to a symmetric tensor theory.

Next, we solve the angular momentum conservation constraint. To do so we express the antisymmetric part of the stress tensor $T_{\text{odd}} = \epsilon^{ij} T_{ij}$ in terms of the tensor electric field

$$T_{\text{odd}} = \epsilon^{ij} E_{ij} = \epsilon^{ij} \dot{A}_{ij} - \epsilon^{ij} \partial_i \Phi_j. \tag{44}$$

In these variables the conservation of angular momentum equation takes form

$$\partial_0(L^0 + \epsilon^{ij} A_{ij}) + \partial_i(L^i + \epsilon^{ij} \Phi_j) = 0. \tag{45}$$

This equation is then solved via introducing an ordinary $U(1)$ gauge field

$$L^0 + \epsilon^{ij} A_{ij} = \epsilon^{ij} \partial_i a_j = b, \qquad L^i + \epsilon^{ij} \Phi_j = \epsilon^{ij}(\partial_i a_0 - \partial_0 a_i) = \epsilon^{ij} e_j, \tag{46}$$

where $b$ and $e_i$ are the magnetic and electric fields correspondingly. It is important to note that the action depends on the non-conserved variables $L^\mu$, which leads to a quite unusual gauge redundancy of the action. Indeed, although not conserved, the components of total angular momentum density and current $L^\mu$ are observable. Therefore, the gauge redundancy is comprised of variations of $A_{ij}$ and $a_\mu$ that leave $L^\mu$ unchanged. Solving (46) for $L^\mu$ we find

$$L^0 = -\epsilon^{ij} A_{ij} + b, \qquad L^i = \epsilon^{ij} e_j - \epsilon^{ij} \Phi_j. \tag{47}$$

These are invariant under the following set of transformations

$$\delta a_\mu = \partial_\mu \lambda \tag{48}$$

$$\delta \Phi_i = \dot{\alpha}_i, \qquad \delta A_{ij} = \partial_j \alpha_i, \qquad \delta a_i = -\alpha_i, \qquad \delta a_0 = 0. \tag{49}$$

These transformations constitute the gauge redundancy of the stress tensor.

The action for the dual gauge fields takes form

$$S = \int dt\, d^2x \left[ \tilde{C}^{-1}_{ijkl} E_{ij} E_{kl} + B_i B^i + \zeta^{-1}(b + \epsilon^{ij} A_{ij})^2 + (e^i - \Phi^i)(e_i - \Phi_i) \right]. \tag{50}$$

The canonical momentum conjugate to $A_{ij}$ is given by (21), while the canonical momentum conjugate to $a_i$ is given by

$$\pi^i = e^i - \Phi^i. \tag{51}$$

We can fix a gauge where $a_i = 0$ by choosing $\alpha^i = \epsilon^{ik}a_k$ and $a_0 = \dot{\lambda}$. In this case the source-free action takes form

$$S = \int dt d^2 x \left[ \tilde{C}_{ijkl}^{-1} E_{ij} E_{kl} + B_i B^i + \zeta^{-1}(\epsilon^{ij} A_{ij})^2 - \Phi^i \Phi_i \right]. \tag{52}$$

This action has two gapless and one gapped mode. The gapped mode is the anti-symmetric part of $A_{ij}$ with the mass being fixed by the stiffness of the local orientation, $\zeta$, whereas the remaining components are gapless.

We could have expected having only two gapless modes from the Goldstone's theorem for non-semi-simple groups. Indeed, the gauge fields describe a dual formulation of the Goldstone modes generated via spontaneous breaking of rotational and translation symmetries. These symmetries are not independent and breaking of rotational symmetry does not lead to the new gapless modes. However, there is a gapped mode that corresponds to the local orientation of the medium.

### 3.3 Defects

In the Cosserat theory a more general set of integrability conditions is possible. Namely, we demand that singularity in $\theta$ exactly cancels singularity in $\varphi$. That is [45]

$$\rho_{\text{disl}}^i = 0, \qquad \rho_{\text{disc}} + \rho_\theta = 0, \tag{53}$$

where $\rho_\theta$ is another defect density specific to the Cosserat theory

$$\rho_\theta = \epsilon^{ij} \partial_i \partial_j \theta. \tag{54}$$

It is natural to combine the disclination defects with the angle defects into rotational defects

$$\rho_{\text{rot}} = \rho_{\text{disc}} + \rho_\theta = \epsilon^{ij} \partial_i \partial_j (\varphi + \theta). \tag{55}$$

These disclination defects have two independent contributions. The functional integral is defined to integrate over all possible configurations of $u^i$ and $\theta$, including the singular ones. Equations (53) state that a globally defined $u_i$ is exists if there are no dislocations and disclinations of $u_i$ are canceled by disclinations of $\theta$.

It may be convenient to express the asymmetric strain tensor in terms of the angles

$$\gamma_{ij} = \frac{1}{2}(\partial_i u_j + \partial_j u_i) + \frac{1}{2}(\partial_i u_j - \partial_j u_i) + \epsilon_{ij}\theta = u_{ij} + \epsilon_{ij}(\varphi + \theta). \tag{56}$$

If we denote $\phi = \varphi + \theta$ the action retains the same form except for the $\tau_i \tau^i$ term. That term takes form

$$\int dt d^2 x \tau_i \tau^i \rightarrow \int dt d^2 x (\partial_i \phi - \partial_i \theta)(\partial^i \phi - \partial^i \theta). \tag{57}$$

Thus, $\theta$ cannot be eliminated from the theory and its singularities must be summed over independently.

Next we turn to the study of defect matter and Gauss laws. To this end, we separate the local displacement and local orientation into regular and singular parts: $u^i \rightarrow u^i_{\text{reg}} + u^i_{\text{sing}}$ and $\theta \rightarrow \theta_{\text{reg}} + \theta_{\text{sing}}$ and integrate over $u^i_{\text{reg}}$ and $\theta_{\text{reg}}$, which leads to the angular momentum conservation constraint. The coupling of $u^i_{\text{sing}}$ and $\theta_{\text{sing}}$ to the gauge fields can be brought to the following form

$$\delta S = \int dt d^2 x \left[ u^i_{\text{sing}} \partial_\mu T^{i\mu} + \theta_{\text{sing}} \left( \partial_\mu L^\mu + \epsilon^{ij} T_{ij} \right) \right] \tag{58}$$

$$= \int dt d^2 x \left[ (\rho^i + 2\epsilon^{ij} \partial_j \theta_{\text{sing}}) \Phi_i + (J^{ij} + 2\dot{\theta}_{\text{sing}} \epsilon^{ij}) A_{ij} + j^\mu a_\mu \right], \tag{59}$$

where

$$j^0 = \rho_\theta = \epsilon^{ik}\partial_k\partial_i\theta_{\text{sing}}, \qquad j^i = \epsilon^{ij}(\partial_i\partial_0 - \partial_0\partial_i)\theta_{\text{sing}}. \tag{60}$$

The total disclination density is given by (55).

The Gauss laws are succinctly formulated in terms of the canonical momenta[2]

$$\partial_i\pi^i = \rho_\theta, \qquad \partial_j\Pi^{ij} = \rho^i - e^i, \tag{61}$$

where we have denoted the density of $\theta$-vortices $\varrho = j^0 = \epsilon^{ij}\partial_i\partial_j\theta$.

One immediate consequence of the Gauss law constraint equations is

$$\partial_i\partial_j\Pi^{ij} = \partial_i\rho^i + \rho_\theta = \rho_{\text{rot}}. \tag{62}$$

This corresponds to the fact that both angle singularities and displacement singularities contribute to disclinations (or to the "scalar charge"). The dipole moment of the joint density is conserved

$$D_i = \int x_i(\partial_j\rho^j + \rho_\theta) = \int \rho^i + \int x^i\rho_\theta = \int \partial(\dots) = 0, \tag{63}$$

however neither $\rho^i$ nor $x^i\varrho$ are separately conserved, but they satisfy a constraint

$$\int \rho^i = -\int x^i\rho_\theta. \tag{64}$$

This is not surprising since it can be seen from (59) that dislocation density now consists of two contributions: $\rho^i_{\text{disl}} - 2\partial^i\theta_{\text{sing}}$.

## 3.4 Restricted motion

We consider a general dual theory coupled to the defect currents $(j^\mu, \rho, J^{ij})$. The gauge invariance of the action reflects the conservation laws

$$\partial_\mu j^\mu = 0, \qquad \partial_0\rho^i + \partial_j J^{ij} = j^i. \tag{65}$$

The second equation implies that the current of $j^0$-charges violates the conservation of $J^{ij}$. Taking the divergence of the second equation we find that the total current of $\theta$ and $\varphi$ defects is conserved

$$\begin{aligned}\partial_0\partial_i\rho^i + \partial_i\partial_j J^{ij} = \partial_i j^i = -\partial_0 j^0 \quad &\Rightarrow \quad \partial_0(\partial_i\rho^i + \rho_\theta) + \partial_i\partial_j J^{ij} = 0\\ &\Rightarrow \quad \partial_0\rho_{\text{rot}} + \partial_i\partial_j J^{ij} = 0,\end{aligned} \tag{66}$$

where we have used the relation between the vector charge, disclination density, $\rho_\theta$ and $\rho_{\text{rot}}$.

The physical content of this relation is that singularities in $\varphi$ and $\theta$ are fractons and cannot freely move around. The dipoles of these singularities still satisfy the glide constraint in view of (62). However, there is no way to probe singularities of $\varphi$ and of $\theta$ separately. Both defects are disclinations and cannot be distinguished at the level of longwave effective theory.

Finally, we note that that the massive variable $\epsilon^{ij}A_{ij}$ can be integrated out from (52). After performing the integration we get back to the dual formulation of the symmetric elasticity theory.

---

[2]These Gauss laws generate the gauge transformations (48)-(49). Indeed, Poisson brackets of these relations with the gauge fields give, for example,

$$\delta a_k = \left\{i\int d^2x(\partial_j E^{ij} + e^j)\lambda_i, a_k\right\} = i\int d^2x\lambda_i\{e^j, a_k\} = \lambda_k.$$

### 3.5 Duality in curved space

It has been previously noted that general symmetric tensor gauge theories are not well-defined on a curved space [26,39]. Indeed, when the background geometry is curved the generalized magnetic field is no longer gauge-invariant, thereby increasing the number of effective degrees of freedom. At the same time elasticity is perfectly well-defined on a curved lattice and defects satisfy similar constraints [53,54]. One may wonder how to reconcile these two facts.

Resolution of this conflict is quite simple: it turns out that the duality map is no longer valid in curved space. Indeed, consider the conservation of the stress tensor constraint in curved space. We have

$$\partial_0(\sqrt{g}P^i) + \partial_i(\sqrt{g}T^{ij}) = \Gamma^i{}_{k,j}T^{kj}, \tag{67}$$

where $\Gamma^i{}_{k,j}dx^j$ is the Christoffel symbol. These equations cannot be solved in terms of the gauge fields as in the previous Section. We leave the detailed investigation of curved space dualities to future work.

## 4 Conclusion

We have constructed a dual theory for Cosserat elasticity. It was found that the Cosserat theory is dual to a theory of general tensor gauge field coupled to an ordinary, non-propagating, $U(1)$ gauge field. The defect matter couples to both gauge fields. The stress tensor of the Cosserat theory is not symmetric, which ultimately leads to the dual description in terms of the non-symmetric tensor gauge theory. The odd part of the stress tensor maps onto the antisymmetric part of the tensor gauge field. The latter is massive and can be integrated out to obtain a low energy theory valid below its mass scale. The resulting low energy theory is a theory of a symmetric tensor gauge field, previously discussed in literature. It would be very interesting to extend the dual formulation of elasticity to other spatially ordered phases such as various types of smectic and nematic order.

## Acknowledgements

We acknowledge discussions with Carlos Hoyos, Sergej Moroz and Vijay Shenoy. In the final stages of this work reference [37] appeared on ArXiv, which overlaps with our results.

**Funding information**   A.G. was supported by the Brown University start up funds. P. S. was supported by the Deutsche Forschungsgemeinschaft via the Leibniz Program.

## A  Inverting the general tensor of elastic moduli

In the most general case, without making any assumptions about the form of elastic tensors, beyond isotropy and rotational invariance the action depends on the matrix of elastic coefficients $C_{ijkl}$. This matrix is a rank–4 tensor, which can be efficiently represented in terms of

the following projectors

$$P^0_{ijkl} = \frac{1}{2}\delta_{ij}\delta_{kl}, \tag{68a}$$

$$P^1_{ijkl} = \frac{1}{2}(\delta_{ik}\delta_{jl} - \delta_{il}\delta_{jk}), \tag{68b}$$

$$P^2_{ijkl} = \frac{1}{2}(\delta_{ik}\delta_{jl} + \delta_{il}\delta_{jk}) - \frac{1}{2}\delta_{ij}\delta_{kl}. \tag{68c}$$

One can check that $P^m_{ijab}P^n_{abkl} = P^m_{ijkl}$ if $m = n$ and zero otherwise. The tensor of elastic coefficients can be decomposed into

$$C_{ijkl} = p_0 P^0_{ijkl} + p_1 P^1_{ijkl} + p_2 P^2_{ijkl}. \tag{69}$$

In this form the elasticity matrix can be inverted using the properties of projectors

$$C^{-1}_{ijkl} = \frac{1}{p_0}P^0_{ijkl} + \frac{1}{p_1}P^1_{ijkl} + \frac{1}{p_2}P^2_{ijkl}. \tag{70}$$

We note that in classical elasticity the above tensor is only partially invertible due to the lack of rotational degrees of freedom. In this case $p_1 = 0$ and the inverse is defined only in the invertible subspace

$$C^{-1}_{ijkl} = \frac{1}{p_0}P^0_{ijkl} + \frac{1}{p_2}P^2_{ijkl}. \tag{71}$$

# B  Stückelberg mechanism

## B.1  Stückelberg mechanism in Proca theory

As an example of the Stückelberg mechnism we discuss the massive Proca lagrangian [55,56]

$$\mathcal{L}_{\text{Proca}} = -\frac{1}{4}F^2_{\mu\nu} - \frac{1}{2}m^2 A_\mu A^\mu. \tag{72}$$

An immediate consequence of the mass term is the breaking of the $U(1)$ gauge symmetry

$$A_\mu \rightarrow A_\mu + \partial_\mu \alpha. \tag{73}$$

The idea behind Stückelberg trick is to restore gauge invariance at the expense of an additional gauge field. To see this we can decompose a vector field in the in the transverse and longitudinal components

$$A_\mu = A^T_\mu + \partial_\mu \tilde{\pi}. \tag{74}$$

This procedure allows one to rewrite the Proca Lagrangian

$$\mathcal{L}_{\text{Proca}} = -\frac{1}{2}(\partial_\mu A^T_\nu)^2 - \frac{1}{2}m^2 (A^T_\mu)^2 - \frac{1}{2}(\partial_\mu \pi)^2, \tag{75}$$

where $\pi = m\tilde{\pi}$. We note that the transverse field satisfies

$$\partial^\mu A^T_\mu = 0, \tag{76}$$

which follows if we take the derivative of equations of motion

$$\partial^\mu F^T_{\mu\nu} + m^2 A^T_\nu = 0. \tag{77}$$

Given the above relation we see that our theory propagates a massive transverse mode and a massless scalar mode. The new Lagrangian is also gauge invariant with respect to the following substitutions

$$A_\mu \rightarrow A^T_\mu - \partial_\mu \chi, \tag{78a}$$

$$\pi \rightarrow \pi + m\chi. \tag{78b}$$

### B.2 Stückelberg mechanism in the dual Cosserat theory

To illustrate the Stückelberg mechnism in the context of Cosserat theory we take the action

$$S = \int dt\, d^2x \Big[ \tilde{C}_{ijkl}^{-1} E_{ij} E_{kl} + B_i B^i + (b + \epsilon^{ij} A_{ij})^2 + (e^i - \Phi^i)(e_i - \Phi_i) \Big], \tag{79}$$

and investigate gauge symmetries. The transformations that leave the action invariant read

$$a_0 \to a_0 - \partial_0 \hat{\pi} \tag{80a}$$

$$a_j \to a_j + \partial_j \hat{\pi} - \chi_j, \tag{80b}$$

$$A_{\mu j} \to A_{\mu j} + \partial_\mu \chi_j, \tag{80c}$$

where $\mu \in \{0, 1, 2\}$. We see that because the theory becomes massive due to the coupling between two gauge fields a Stückelberg $\hat{\pi}$ naturally appears.

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
