# Peer review of "On duality between Cosserat elasticity and fractons"

_SciPost Physics, doi:SciPost Phys. 8, 065 (2020)_

## Round 1 · Referee Report · Anonymous (Referee 1) · 2019-11-22

Report

This manuscript formulates and discusses a duality for the so-called Cosserat theory of elasticity. To set the stage, the authors first review the duality between ordinary (i.e. symmetric) two-dimensional elasticity theory and a tensor gauge theory for fractons, which has been discussed in recent literature. Then, following the same lines, they formulate the duality transformation for the case of Cosserat elasticity. They find that the dual theory is a non-symmetric tensor tensor gauge theory with one additional massive degree of freedom with respect to the symmetric case.

The paper is well organised and well written. The results are sound and interesting and the topic is timely. I think the paper deserves publication. I do have some comments, though, that the authors should consider before publication.

1- Conventions: the authors do not seem to follow usual conventions (see, e.g., Landau book): they omit a factor $1/2$ in the definition of the strain tensor and a factor $1/2$ in the definition of the elastic action. Is there a special reason for that? If not, I would encourage them to use the standard conventions.

2- It should be specified at the beginning of section 2 that it is assumed that the elastic action is isotropic. What happens to the dual theory if the elastic action is not isotropic?

3- In Sec. 2 it would be useful to have the expression of the stress tensor in terms of the strain tensor.

4- In Eq. 4, the inverse of the elastic moduli tensor appear. In the appendix A we are told that this tensor is only partially invertible. Then, what is the precise meaning of the inverse C appearing in Eq. 4?

5- In Eq. 5 the effective action, after integration over the smooth part of the displacement field, still depends on its singular part. The notation should reflect this.

6- In Eq. 11 the symbol $\xi^i$ is used, which appears also in Eq. 35 (without index) with a completely different meaning. This should be avoided.

7- It would be useful to extend the introduction of the section on Cosserat elasticity theory. This theory is probably unfamiliar to most readers and the authors should explain a bit more extensively why it is physically interesting and relevant. For example, they could mention concrete examples of materials whose elastic properties are described by this theory.

8- In Eq. 35 the symbol $\xi$ should be defined.

9- I am a bit confused by the form of the action in Eq. 35. If, as usual, the tensor of elastic moduli $C_{ijkl}$ is symmetric under exchange of i and j (and of k and l), the term $\epsilon_{ij}\theta$ in the strain tensor $\gamma_{ij}$ does not contribute to the action term $C_{ijkl}\gamma_{ij}\gamma_{kl}$ and as consequence the field $\theta$ should be decoupled from the field $u_i$. If, on the other side, the tensor $C$ is not symmetric, then in the limit $\theta=0$, the action would depend on both the symmetric and antisymmetric part of the strain tensor, so the theory would not reduce to the symmetric one for $\theta=0$. The authors should clarify this point.

10- This is related to point 7: At the beginning of section 3.3, the authors introduce the "integrability conditions" in the Cosserat theory. The physical explanation and justification should be given here.

11- This is also related to point 7: While it is well-known from classical elasticity theory that singular configurations of $u_i$ are related to disclinations, it is much less know what is the physical meaning of a singular configuration of the field $\theta$. When introducing it, the authors should briefly explain what is it.

---

## Round 2 · List of Changes

\documentclass{article}
\usepackage{graphics}
\usepackage{amssymb}
\usepackage{amsmath}
\usepackage{graphicx}
\usepackage{color}
\setlength{\textwidth}{17.75cm}
\setlength{\textheight}{23.0cm}
\setlength{\topmargin}{-2.cm}
\setlength{\oddsidemargin}{-0.75cm}
\begin{document}
\noindent
Dear Editor,
\\
\noindent
We thank you for considering our revised manuscript for publication in SciPost. We also thank the Referee for taking their valuable time to write the report on our paper. Below we address the points raised by the Referee, and make appropriate changes in the revised manuscript.
\\
\medskip
\medskip
\medskip
\medskip
\emph{$1$-- Conventions: the authors do not seem to follow usual conventions (see, e.g., Landau book):
they omit a factor $1/2$ in the definition of the strain tensor and a factor $1/2$ in the definition of
the elastic action. Is there a special reason for that? If not, I would encourage them to use the
standard conventions.}
\medskip
\medskip
\medskip
\medskip
We chose the convention for symmetric elasticity to facilitate the comparison with Cosserat elasticity. The conventions form Landau's book are in line with classical elasticity, where symmetrization produces factors 1/2. We added a comment on this in the manuscript.
\medskip
\medskip
\medskip
\medskip
\emph{$2$-- It should be specified at the beginning of section 2 that it is assumed that the elastic action
is isotropic. What happens to the dual theory if the elastic action is not isotropic?}
\medskip
\medskip
\medskip
\medskip
To the leading order in gradients breaking $SO(2)$ down to $C_n$ with $n\geq 3$ leads to the same action. Consequently there is no difference \emph{to the leading order in gradients}. To the sub-leading orders more terms will be allowed by $C_n$ than by $SO(2)$. Consequently angular momentum will not be conserved leading to complications with the gauge invariance. Understanding the duality to higher orders in gradients for $C_n$ is an open problem to the best of our knowledge. Most likely the gauge field $A^i_\mu dx^\mu$ will live in vector representation of $C_n$.
If the $SO(2)$ is broken down to $C_2$ then more elastic constants are possible to the \emph{leading} order in gradients. This will be reflected in the form of $C_{ijkl}$ which will be a $C_2$ invariant tensor and will, consequently, include not just $\delta_{ij}$, but also Pauli matrices $\sigma^1_{ij}$ and $\sigma^3_{ij}$. The number of degrees freedom will be the same as in the isotropic theory, but dispersion relation will be anisotropic. The dual gauge theory explicitly depends on $C_{ijkl}$, and consequently will include new anisotropic contractions between components of the tensor electric field. If we include the angular degree of freedom then there is again an issue with angular momentum conservation and we leave it as an open problem.
If $SO(2)$ is broken down completely then initial elasticity theory is quite exotic as it cannot come from a UV model defined on a Bravais lattice. We do not have anything to say regarding this case.
\medskip
\medskip
\medskip
\medskip
\emph{$3$ -- In Sec. 2 it would be useful to have the expression of the stress tensor in terms of the strain
tensor.}
\medskip
\medskip
\medskip
\medskip
The expression has been added.
\medskip
\medskip
\medskip
\medskip
\emph{$4$-- In Eq. 4, the inverse of the elastic moduli tensor appear. In the appendix A we are told that
this tensor is only partially invertible. Then, what is the precise meaning of the inverse C
appearing in Eq. 4?}
\medskip
\medskip
\medskip
\medskip
Inversion $C_{ijkl}$ is explained in the Appendix A. We have added a sentence referring the reader to the Appendix. It is invertible on the subspace of symmetric tensors. ($C_{ijkl}$ acts on rank-$2$ tensors. We consider a subspace of symmetric tensors, which is closed under addition and consequently is a proper subspace. On this subspace $C_{ijkl}$ is invertible. The procedure explaining how to invert it on this subspace is described in the Appendix A).
At the end of the day we are computing a Gaussian integral. It can be calculated by turning $u_{ij}$ into a vector $u_I$ and $C_{ijkl}$ into a symmetric matrix $C_{IJ}$. The resulting gaussian integral is non-degenerate and its computation reduces to inverting $C$.
\medskip
\medskip
\medskip
\medskip
\emph{$5$-- In Eq. 5 the effective action, after integration over the smooth part of the displacement field,
still depends on its singular part. The notation should reflect this.}
\medskip
\medskip
\medskip
\medskip
We modified the expression for the effective action to emphasise explicit dependence on the singular part of the displacement vector.
\medskip
\medskip
\medskip
\medskip
\emph{$6$-- In Eq. 11 the symbol is used, which appears also in Eq. 35 (without index) with a
completely different meaning. This should be avoided.}
\medskip
\medskip
\medskip
\medskip
We thank the referee for pointing this out. We kept $\xi ^i$ as a vector generating diffeomorphisms and changed the elastic coefficient to $\zeta$.
\medskip
\medskip
\medskip
\medskip
\emph{$7$-- It would be useful to extend the introduction of the section on Cosserat elasticity theory. This
theory is probably unfamiliar to most readers and the authors should explain a bit more
extensively why it is physically interesting and relevant. For example, they could mention
concrete examples of materials whose elastic properties are described by this theory.}
\medskip
\medskip
\medskip
\medskip
Following the referee's suggestion we added references with examples of metamaterials where the effects of Cosserat elasticity are important and have been observed.
We also note that the duality we have presented is applicable to ordinary elasticity, which emerges at the energies lower than the gap of $a_\mu$.
\medskip
\medskip
\medskip
\medskip
\emph{$8$-- In Eq. 35 the symbol should be defined.}
\medskip
\medskip
\medskip
\medskip
$\zeta$ is an additional elastic \textbf{defined} by Eq. 35. It characterizes the generalized elasticity of the material: there is a Hooke's law for deviation of the local orientation $\theta$ from a constant equilibrium value.
\medskip
\medskip
\medskip
\medskip
\emph{$9$-- I am a bit confused by the form of the action in Eq. 35. If, as usual, the tensor of elastic
moduli $C_{ijkl}$ is symmetric under exchange of i and j (and of k and l), the term $\epsilon_{ij} \theta$ in the strain $\gamma_{ij}$
tensor does not contribute to the action term $C_{ijkl} \gamma_{ij}\gamma_{kl}$ and as consequence the field
should be decoupled from the field . If, on the other side, the tensor is not symmetric, then
in the limit $\theta=0$ , the action would depend on both the symmetric and antisymmetric part of
the strain tensor, so the theory would not reduce to the symmetric one for $\theta=0$. The authors should clarify this point. }
\medskip
\medskip
\medskip
\medskip
Upon setting $\theta=0$ the action depends on the antisymmetric part of the strain tensor, which is equal to $\epsilon_{ij} \varphi$. Under global rotations by an angle $\theta$ the variable $\varphi$ transforms as $ \varphi^\prime = \varphi+ \theta$. Then using the requirement of global rotational invariance we can simply choose a frame where $ \varphi=0$. This reduces the free energy $\int C_{ijkl} \gamma_{ij}\gamma_{kl}$ to the standard form $\int C_{ijkl} u_{ij}u_{kl}$. The anti-symmetric components of $C_{ijkl}$ automatically drop out.
\medskip
\medskip
\medskip
\medskip
\emph{$10$-- This is related to point 7: At the beginning of section 3.3, the authors introduce the "integrability conditions" in the Cosserat theory. The physical explanation and justification should be given here.}
\medskip
\emph{$11$-- This is also related to point 7: While it is well-known from classical elasticity theory that singular configurations of $u_i$
are related to disclinations, it is much less know what is the physical meaning of a singular configuration of the field $\theta$. When introducing it, the authors should briefly explain what is it.}
\medskip
\medskip
\medskip
\medskip
We provide a single response to both questions.
\medskip
We have added a sentence clarifying the meaning of integrability conditions, below Eq.(55). Integrability conditions refer to the conditions for existence of smooth, globally defined displacement vector. These conditions are equivalent to the absence of lattice defects. In Cosserat theory we have a slightly more general case, when both $\vec{u}$ and $\theta$ are singular, but their disclination singularities cancel exactly.
The singularities of $\theta$ and of $\varphi = \vec{\partial} \times \vec{u}$ have the same meaning -- these are defects of rotational symmetry or disclinations. At the level of effective theory we cannot distinguish these two defects. We can only state with certainty that there is a disclination in the medium because parallel transport of a vector along a closed loop containing a disclination leads to a rotation.
We have provided more explicit formulas for dislocation and disclination densities below Eqs. (58-59).
The singularities in $\theta$ have a very elegant interpretation in the geometric language. In geometric formulation of elasticity disclinations map to curvature and dislocations to torsion. From this point of view ordinary elasticity corresponds to geometry with torsion and Levi-Civita connection $(T,\omega_{\rm LC})$, while Cosserat elasticity correspond to a geometry with torsion and a \emph{general} connection $(T, \omega)$. The latter can always be decomposed into Levi-Civita part and the rest $\omega = \omega_{LC} + (\omega - \omega_{LC})$. Corresponding curvature has, then, two contributions: one coming from $\omega_{LC}$ and one coming from the contorsion tensor $K =\omega - \omega_{LC}$. These two contributions cannot be distinguished by parallel transport experiment.
\medskip
\medskip
\medskip
\medskip
\noindent
We hope that you will kindly consider the resubmitted manuscript for publication in SciPost. We are looking forward to your kind consideration.
\\
\noindent
Sincerely yours
\\
\noindent
A. Gromov, P. Sur\'{o}wka
\end{document}
\usepackage{graphics}
\usepackage{amssymb}
\usepackage{amsmath}
\usepackage{graphicx}
\usepackage{color}
\setlength{\textwidth}{17.75cm}
\setlength{\textheight}{23.0cm}
\setlength{\topmargin}{-2.cm}
\setlength{\oddsidemargin}{-0.75cm}
\begin{document}
\noindent
Dear Editor,
\\
\noindent
We thank you for considering our revised manuscript for publication in SciPost. We also thank the Referee for taking their valuable time to write the report on our paper. Below we address the points raised by the Referee, and make appropriate changes in the revised manuscript.
\\
\medskip
\medskip
\medskip
\medskip
\emph{$1$-- Conventions: the authors do not seem to follow usual conventions (see, e.g., Landau book):
they omit a factor $1/2$ in the definition of the strain tensor and a factor $1/2$ in the definition of
the elastic action. Is there a special reason for that? If not, I would encourage them to use the
standard conventions.}
\medskip
\medskip
\medskip
\medskip
We chose the convention for symmetric elasticity to facilitate the comparison with Cosserat elasticity. The conventions form Landau's book are in line with classical elasticity, where symmetrization produces factors 1/2. We added a comment on this in the manuscript.
\medskip
\medskip
\medskip
\medskip
\emph{$2$-- It should be specified at the beginning of section 2 that it is assumed that the elastic action
is isotropic. What happens to the dual theory if the elastic action is not isotropic?}
\medskip
\medskip
\medskip
\medskip
To the leading order in gradients breaking $SO(2)$ down to $C_n$ with $n\geq 3$ leads to the same action. Consequently there is no difference \emph{to the leading order in gradients}. To the sub-leading orders more terms will be allowed by $C_n$ than by $SO(2)$. Consequently angular momentum will not be conserved leading to complications with the gauge invariance. Understanding the duality to higher orders in gradients for $C_n$ is an open problem to the best of our knowledge. Most likely the gauge field $A^i_\mu dx^\mu$ will live in vector representation of $C_n$.
If the $SO(2)$ is broken down to $C_2$ then more elastic constants are possible to the \emph{leading} order in gradients. This will be reflected in the form of $C_{ijkl}$ which will be a $C_2$ invariant tensor and will, consequently, include not just $\delta_{ij}$, but also Pauli matrices $\sigma^1_{ij}$ and $\sigma^3_{ij}$. The number of degrees freedom will be the same as in the isotropic theory, but dispersion relation will be anisotropic. The dual gauge theory explicitly depends on $C_{ijkl}$, and consequently will include new anisotropic contractions between components of the tensor electric field. If we include the angular degree of freedom then there is again an issue with angular momentum conservation and we leave it as an open problem.
If $SO(2)$ is broken down completely then initial elasticity theory is quite exotic as it cannot come from a UV model defined on a Bravais lattice. We do not have anything to say regarding this case.
\medskip
\medskip
\medskip
\medskip
\emph{$3$ -- In Sec. 2 it would be useful to have the expression of the stress tensor in terms of the strain
tensor.}
\medskip
\medskip
\medskip
\medskip
The expression has been added.
\medskip
\medskip
\medskip
\medskip
\emph{$4$-- In Eq. 4, the inverse of the elastic moduli tensor appear. In the appendix A we are told that
this tensor is only partially invertible. Then, what is the precise meaning of the inverse C
appearing in Eq. 4?}
\medskip
\medskip
\medskip
\medskip
Inversion $C_{ijkl}$ is explained in the Appendix A. We have added a sentence referring the reader to the Appendix. It is invertible on the subspace of symmetric tensors. ($C_{ijkl}$ acts on rank-$2$ tensors. We consider a subspace of symmetric tensors, which is closed under addition and consequently is a proper subspace. On this subspace $C_{ijkl}$ is invertible. The procedure explaining how to invert it on this subspace is described in the Appendix A).
At the end of the day we are computing a Gaussian integral. It can be calculated by turning $u_{ij}$ into a vector $u_I$ and $C_{ijkl}$ into a symmetric matrix $C_{IJ}$. The resulting gaussian integral is non-degenerate and its computation reduces to inverting $C$.
\medskip
\medskip
\medskip
\medskip
\emph{$5$-- In Eq. 5 the effective action, after integration over the smooth part of the displacement field,
still depends on its singular part. The notation should reflect this.}
\medskip
\medskip
\medskip
\medskip
We modified the expression for the effective action to emphasise explicit dependence on the singular part of the displacement vector.
\medskip
\medskip
\medskip
\medskip
\emph{$6$-- In Eq. 11 the symbol is used, which appears also in Eq. 35 (without index) with a
completely different meaning. This should be avoided.}
\medskip
\medskip
\medskip
\medskip
We thank the referee for pointing this out. We kept $\xi ^i$ as a vector generating diffeomorphisms and changed the elastic coefficient to $\zeta$.
\medskip
\medskip
\medskip
\medskip
\emph{$7$-- It would be useful to extend the introduction of the section on Cosserat elasticity theory. This
theory is probably unfamiliar to most readers and the authors should explain a bit more
extensively why it is physically interesting and relevant. For example, they could mention
concrete examples of materials whose elastic properties are described by this theory.}
\medskip
\medskip
\medskip
\medskip
Following the referee's suggestion we added references with examples of metamaterials where the effects of Cosserat elasticity are important and have been observed.
We also note that the duality we have presented is applicable to ordinary elasticity, which emerges at the energies lower than the gap of $a_\mu$.
\medskip
\medskip
\medskip
\medskip
\emph{$8$-- In Eq. 35 the symbol should be defined.}
\medskip
\medskip
\medskip
\medskip
$\zeta$ is an additional elastic \textbf{defined} by Eq. 35. It characterizes the generalized elasticity of the material: there is a Hooke's law for deviation of the local orientation $\theta$ from a constant equilibrium value.
\medskip
\medskip
\medskip
\medskip
\emph{$9$-- I am a bit confused by the form of the action in Eq. 35. If, as usual, the tensor of elastic
moduli $C_{ijkl}$ is symmetric under exchange of i and j (and of k and l), the term $\epsilon_{ij} \theta$ in the strain $\gamma_{ij}$
tensor does not contribute to the action term $C_{ijkl} \gamma_{ij}\gamma_{kl}$ and as consequence the field
should be decoupled from the field . If, on the other side, the tensor is not symmetric, then
in the limit $\theta=0$ , the action would depend on both the symmetric and antisymmetric part of
the strain tensor, so the theory would not reduce to the symmetric one for $\theta=0$. The authors should clarify this point. }
\medskip
\medskip
\medskip
\medskip
Upon setting $\theta=0$ the action depends on the antisymmetric part of the strain tensor, which is equal to $\epsilon_{ij} \varphi$. Under global rotations by an angle $\theta$ the variable $\varphi$ transforms as $ \varphi^\prime = \varphi+ \theta$. Then using the requirement of global rotational invariance we can simply choose a frame where $ \varphi=0$. This reduces the free energy $\int C_{ijkl} \gamma_{ij}\gamma_{kl}$ to the standard form $\int C_{ijkl} u_{ij}u_{kl}$. The anti-symmetric components of $C_{ijkl}$ automatically drop out.
\medskip
\medskip
\medskip
\medskip
\emph{$10$-- This is related to point 7: At the beginning of section 3.3, the authors introduce the "integrability conditions" in the Cosserat theory. The physical explanation and justification should be given here.}
\medskip
\emph{$11$-- This is also related to point 7: While it is well-known from classical elasticity theory that singular configurations of $u_i$
are related to disclinations, it is much less know what is the physical meaning of a singular configuration of the field $\theta$. When introducing it, the authors should briefly explain what is it.}
\medskip
\medskip
\medskip
\medskip
We provide a single response to both questions.
\medskip
We have added a sentence clarifying the meaning of integrability conditions, below Eq.(55). Integrability conditions refer to the conditions for existence of smooth, globally defined displacement vector. These conditions are equivalent to the absence of lattice defects. In Cosserat theory we have a slightly more general case, when both $\vec{u}$ and $\theta$ are singular, but their disclination singularities cancel exactly.
The singularities of $\theta$ and of $\varphi = \vec{\partial} \times \vec{u}$ have the same meaning -- these are defects of rotational symmetry or disclinations. At the level of effective theory we cannot distinguish these two defects. We can only state with certainty that there is a disclination in the medium because parallel transport of a vector along a closed loop containing a disclination leads to a rotation.
We have provided more explicit formulas for dislocation and disclination densities below Eqs. (58-59).
The singularities in $\theta$ have a very elegant interpretation in the geometric language. In geometric formulation of elasticity disclinations map to curvature and dislocations to torsion. From this point of view ordinary elasticity corresponds to geometry with torsion and Levi-Civita connection $(T,\omega_{\rm LC})$, while Cosserat elasticity correspond to a geometry with torsion and a \emph{general} connection $(T, \omega)$. The latter can always be decomposed into Levi-Civita part and the rest $\omega = \omega_{LC} + (\omega - \omega_{LC})$. Corresponding curvature has, then, two contributions: one coming from $\omega_{LC}$ and one coming from the contorsion tensor $K =\omega - \omega_{LC}$. These two contributions cannot be distinguished by parallel transport experiment.
\medskip
\medskip
\medskip
\medskip
\noindent
We hope that you will kindly consider the resubmitted manuscript for publication in SciPost. We are looking forward to your kind consideration.
\\
\noindent
Sincerely yours
\\
\noindent
A. Gromov, P. Sur\'{o}wka
\end{document}

---

## Editorial Decision

published